# Urban Rural Interaction: Processes and Changes in the *Marina Oriental* of Cantabria (Spain)

Sara Lagüera Díaz

Department of Geography, Urban and Regional Planning, University of Cantabria, 39005 Santander, Spain;
saralaguera@hotmail.com

**Abstract:** Since the middle of the last century, especially since the seventies, processes have been generated and consolidated that have changed the image of certain rural environments in Spain, especially coastal, with new forms of organization and territorialities that break the traditional model. The Cantabrian territory, like other areas of the Spanish coast, has seen its territories and landscapes altered in terms of its demographic, economic, and urban structures. The variation over the easternmost area of the Autonomous Community of Cantabria is significant, affected by various growth processes of both cities in the region, as well as others adjacent and connected, such as the urban agglomeration of Bilbao, influencing this space that we call *Marina Oriental* de Cantabria. The justification and objectives are to know how the coastal geographical situation, good communications, and proximity to Bilbao have configured this space to become a functional part of the metropolitan agglomeration that is generated around this city. An investigation focused on the analysis of the intensification and the effects of the urbanization process of a rural and rururban area, from an integrative, transversal, and multiscale approach, supported by inductive and hybrid methodology, with quantitative and qualitative methods. Through this study, the evolution and problems of these spaces will be known, to analyze the processes that have occurred and continue to occur and, thus, propose measures to reduce the negative effects. The main results and conclusions of the research are manifested in transformations on a legacy space, which has been productively redefined, being one of the most changed since the middle of the last century.

**Keywords:** rural–urban interaction; deagrarianisation; economic tertiarization; urban expansion; land use



## 1. Introduction

The urbanization processes that developed in Spain have been manifesting themselves more intensively on the territory since the nineteenth century and especially during the second half of the twentieth century, with characteristics that have been very similar to those of other countries, although at a different scale and pace.

These changes and transformations of urbanization, therefore, have not been an isolated case, but have been widespread, in most developed countries, especially in the last fifty years. A clear example is found in the United States, with a strong suburbanization after World War II. In Europe, cities have been more compact than in the American case, but their transformation also accelerated during the second half of the twentieth century, with a suburban trend, the rise of communication and transport networks, with very important environmental, social, and economic consequences, often irreversible [1].

Another clear example at the European level is found in Italy, where in recent years there has been a process of strong anthropic pressure that has generated the change of 90% of the coastal environment, leaving intact only 10% of the original habitat, where 30% of Italians also live [2].

This whole process has also been interesting in other countries outside the European Union, such as the case of China [3], where the urbanization process was promoted from

the nineties onwards, with a great restructuring and interaction between rural and urban areas, with new rural economies and tourism, reflecting the change in land use, life and, ultimately, the economy [4].

In the Spanish case, this rapid and abrupt growth of urbanization processes has a great impact on highly vulnerable spaces, such as coastal rural areas [5,6], which makes it necessary to develop in-depth and extensive studies that reflect on it in an extensive way, in a context and a moment, in addition, of great importance and social involvement in environmental and sustainability issues. These spaces are the most suitable and attractive for construction, with a high growth rate and a boom in tourism and the artificialization of the territory [7].

It has more importance, if possible, within the *Marina Oriental*, a territory that has undergone enormous transformations both socially, economically, and urbanistically in a relatively short period of time, with a practical shortage of adequate planning and planning figures and updated to the requirements of each moment. In the processes of urbanization many factors such as social, demographic, economic and territorial, are involved. All of them come very close to the factors, closer and developed over time, that have a lot to do with the increase and improvement in mobility in recent years.

Thanks to the improvement of accessibility, transport infrastructures, access to the car, and, in general, the improvement of the standard of living, work movements have increased both in number and in travel distances. Thus, the population is no longer looking for such proximity to workplaces but is increasingly opting for places of residence relatively far from urban agglomerations, which are mainly those with the largest job market.

With this increase in mobility, it is possible to cover wider spaces thanks to the existing good connection, creating new residential environments outside urban agglomerations, in spaces previously dedicated to other uses and with other functions. These spaces, in addition, have become highly attractive for the population looking for a life outside the city but at the same time a good connection with it, due to their dependence for different reasons. Therefore, it has connected and brought the urban environment to the rural environment in a very remarkable way, restructuring many of its land uses and other characteristic features and generating an expansion of residential functions, in this case, especially, of a vacation and secondary nature, such as second homes, with a clear prominence of the area of the *Marina Oriental* of Cantabria.

If we analyze the evolution of cities, starting from the twentieth century, periods of stagnation and urban growth can be distinguished, which happened over time with different intensity. However, practically all of them have had a strong growth trend since the nineties until the period prior to the economic crisis of the early XXI century, when they went into an economic decline, generating a strong slowdown of the urbanization process that had been achieved [8,9] and that had marked the change in image of many Spanish territories.

It is during this period that Spanish cities underwent the most significant changes in terms of their structure and morphology through a dynamic in which new spaces and metropolitan areas are consolidated based on a diffuse urbanization model, making the task of differentiation and the boundary between urban and rural increasingly complex [10,11]. In relation to this process, there is talk of a new concept of urbanization, whose fundamental pillar is related to the dispersion of the central city, which some authors describe as "neourbanism", "new urbanism", emerged around the XXI century society, to differentiate it from the urbanism that emerged in previous eras. To this end, ten principles of what this new urbanism would be are established, oriented to respond to current urban needs. This process is characterized by the change in cities, which have been modernizing, adapting to new needs, and emerging new processes, such as "metapolization" [12], a new term that encompasses metropolization and the consequent formation of spaces that function independently, exceeding the limits of cities and occupying more and more rural spaces around them.

The first to realize that these processes were taking place were the American scientists, in the decade of the seventies, specifically in the field of demography, appreciating that there was an important movement of people who moved from the countryside to the city and vice versa, a movement that emerged especially around the search for work and better living conditions. In fact, they described a city–country migratory movement and pattern, partially contrary to that given in previous decades, which they called "counterurbanization" [13], with migratory flows in the opposite direction to those of previous ur-banization [14–16].

Little by little, due to the importance and transcendence that they were having in other countries, these dynamics were attracting the attention of many other experts. In this way, in recent decades they have spread to various territories, being the object of study in many countries and areas even on a smaller scale.

It is also necessary to consider the importance of the urban process on the environment and animals. The expansion of human settlement has also affected these media, conditioning their distribution, density, movement, etc., since changes in soil conditions are important for both landscapes and living beings. For this, an important tool, especially to visualize all these changes, is geovisualization, a tool that combines the tradition of cartography and geography but integrates the representation and analysis of data, which are useful for many everyday uses.

It is also important, in this introduction, to delimit the area of study that will be worked on and has already been cited previously, the *Marina Oriental* of the Autonomous Community of Cantabria, which, on a small scale, is a paradigmatic example of this process. The various municipalities that make up this territory have experienced remarkable growth dynamics because of the decentralization and dispersion of the great city of Bilbao, located in the Basque Autonomous Community, and, to a lesser extent, of the regional capital, Santander. A process, therefore, that has occurred above the political–administrative limits, which do not seem to have affected the functional configuration of this space, but at the internal level of the administrations, both local and autonomous, specifically in the matter of urbanism and endowments [17].

Therefore, the central objective of this research is to demonstrate the importance of this space, undoubtedly, a large part of which is functionally integrated into the metropolitan area of Bilbao, with numerous changes in its social, economic, urban, and ultimately territorial and landscape structure, in a short and intense time. Therefore, the process of expansion in the urban areas of Bilbao and Santander should be considered the engine of the changes experienced in the Marina Oriental.

To do this, we will start from the central hypothesis, which will be discussed and verified throughout the document, based on the fact that, since the middle of the last century, Cantabria, and especially the *Marina Oriental*, has been intensely affected by urbanization processes, more specifically of perimetropolization, starting from the growth and urban dispersion of Bilbao mainly and of the regional capital, Santander, secondly, giving rise to a series of important territorial changes and impacts, especially de-urbanization and tourist tertiary, as well as population and real estate growth, which entails a partial territorial disarticulation.

It is important to frame the research and its importance when answering a series of questions that arise here. As mentioned in the hypothesis, an urbanization process that has taken place very intensively on a territory that has undergone major transformations in a very short time will be analyzed. This process brings with it a series of problems and deficits, which in many cases are practically impossible to solve, so this research aims to answer all this.

Recent economic, demographic, or social changes, among others, have profoundly modified rural and urban areas, thereby generating the appearance of new types of territories characterized by different degrees of urbanization. Hence the large number of techniques and typologies of the classification and cataloguing of these spaces that exist and that have emerged especially in more recent times [18].

Defining and differentiating rural and urban spaces has become a complex task. The concept of rural has been changing during the last decades from being a synonym of backwardness and only linked to agrarian activity to having more and more positive meanings and being associated with quality of life, with much more variety of uses, economic activities, and functions than it had in the past [19,20]. It has gone from rural spaces whose only function was essentially the production of food and raw materials to something more plural and multifunctional, defined by many authors with the expression "new rurality"[1]. This concept is being widely used in recent times to describe the novel forms of organization and change of functions in non-urban spaces.

What we see today in most rural areas is a progressive de-dollarization and tertiarization of the economy, where agriculture is progressively losing weight in favor of other activities; in fact, it currently occupies just over 4% of the total assets in Spain, according to the latest active population survey, consolidating with it a rural pluriactivity. For its part, outsourcing has been gaining weight, and today it occupy to a very high percentages of assets; specifically, according to the latest data referred to in the previous source, it covers about 70% of the economy. On the other hand, the rest of the sectors, such as industry and construction, dealing 12% and 6% of the assets, respectively.

Another remarkable fact is the so-called "urban exodus" or counterurbanization, a process by which some types of rural spaces have been recovering the population that little by little they were losing during the main period of the "rural exodus" started in the sixties and that has continued for several decades.

## 2. Study Area

The *Marina Oriental* is located between the southern area of the Santander Bay arch and the border with the province of Vizcaya. It is 60 km long and 477.12 square kilometers, that is, almost 9% of the total area of the Autonomous Community of Cantabria. Despite its limited dimensions, it is one of the most developed areas and has the greatest population and economic weight within the region.

The study area consists of a total of seventeen municipalities, from east to west: Castro Urdiales, Liendo, Guriezo, Laredo, Colindres, Limpias, Bárcena de Cicero, Santoña, Argoños, Escalante, Noja, Arnuero, Meruelo, Bareyo, Ribamontán al Mar, Marina de Cudeyo, and Medio Cudeyo (Figure 1).

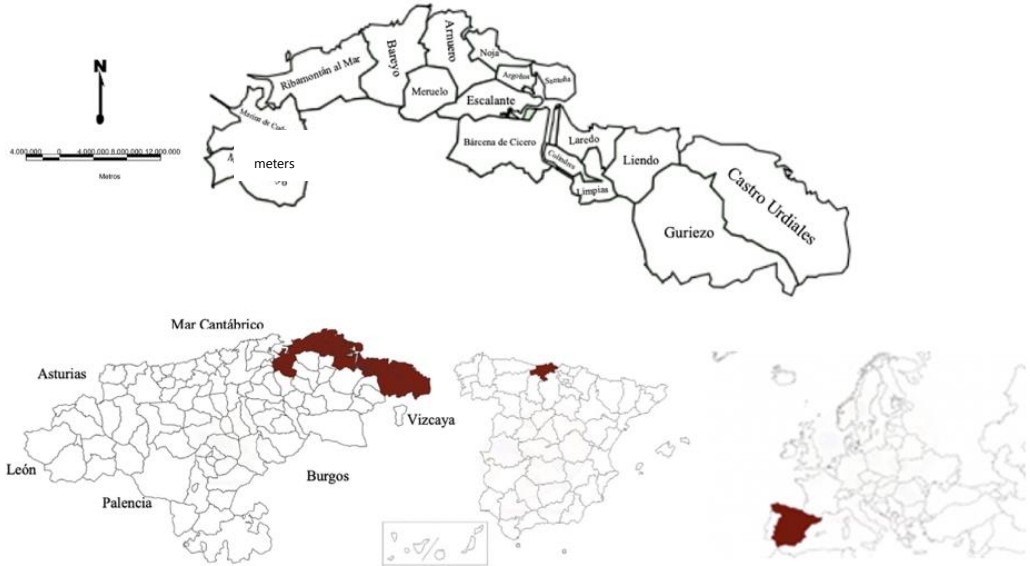

**Figure 1.** Study area of the *Marina Oriental* of Cantabria. Source: own elaboration based on the cartography of the National Geographic Institute (IGN) and the Government of Cantabria.

Several common attributes that define and differentiate this space from the *Marina Oriental*. The first of these is the characteristic relief, which in turn will define the coastal landscape of the environment. The set of these municipalities is not long than ten kilometers from the coastline, influenced by the dynamics and coastal ecosystems. The sea is therefore a key and valuable natural element that has directly influenced the human being and with it the territories on which it inhabits, both positively and negatively.

In fact, this area can be considered the part of Cantabria, along with the rest of the coast of the region, more anthropized and pressured by human action. However, it is at the same time a territory with a high landscape and environmental value that, on occasions, has been kept practically intact until relatively recently.

This territorial area also stands out due to the great tourist reception it produces every year and that is especially concentrated during the central summer months and festive periods. The reasons for this attraction are mainly the tourist and landscape resources it has, focused especially on the numerous beaches and the attractive landscape, one of the references and elements of greater value of the study area. In addition, it has a wide range of both hotel and non-hotel accommodation and secondary housing [21,22].

To this is added a dense and complete communication network and road network (Figure 2) governed by the axis of the Cantabrian Highway A-8, which allows fast and effective communication with the Basque and Asturian Autonomous Communities, as well as with the plateau highway A-67, which connects inland with the Autonomous Community of Castilla y León.

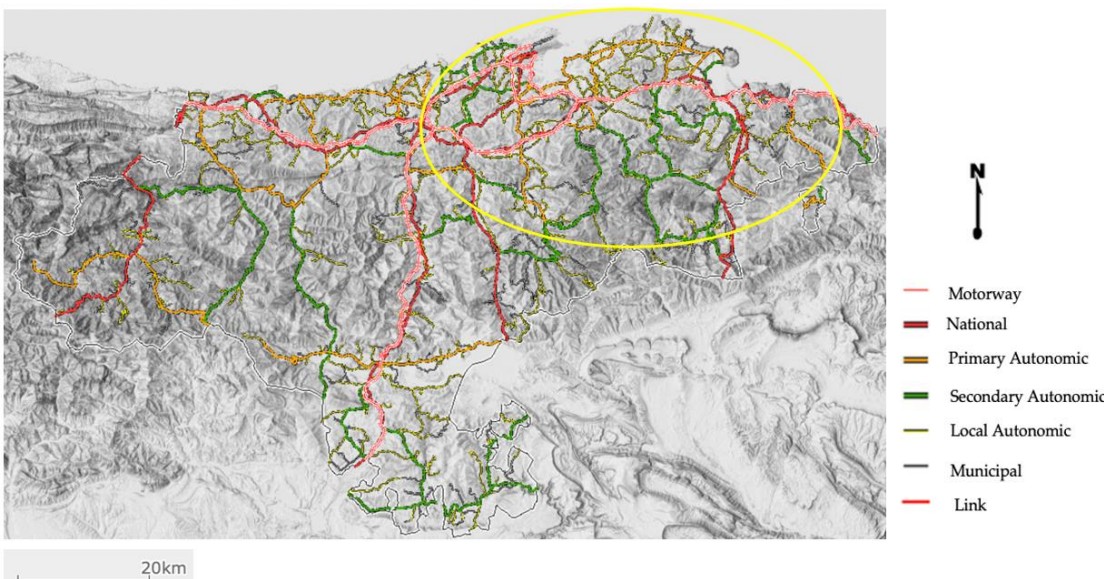

**Figure 2.** Network of Cantabria on relief map. Source: Ministry of Public Works, Spatial Planning and Urban Planning of the Government of Cantabria. Digital model of the terrain.

## 3. Materials and Methods

The methodology applied in this research is of an inductive and hybrid nature, with quantitative and qualitative methods, based on the study of a series of indicators that have allowed the verification of the urbanization processes that have occurred in the space of the *Marina Oriental* of Cantabria, all with the completion of the essential fieldwork. This fieldwork has been carried out visiting all the municipalities of the study area, analyzing the current social, economic, and urban situation, to understand the changes that have developed over time. Photos have been taken, aerial images through drone; data have been obtained from different municipal agencies, the locals, etc.; in short, an attempt has been made to exploit each territory as much as possible.

Quantitative indicators have also been used to measure urbanization processes in other territories and scales. Its application is essentially at the local level. These indicators are mainly of a socio-demographic, economic, urban, and real estate type, addressed within a time ranging from the middle of the last century to the most recent data of each analysis variable.

These statistical data will be obtained mainly from sources generated by the National Institute of Statistics (INE) through its censuses and annual registers. Data such as the summer population figure or daily movements have been obtained in most cases through estimates. However, they are tools that, although their information is not error-free, are the most reliable and with the data available to the study territory.

Regarding socio-demographic indicators, the evolution of the population has been analyzed through the censuses and registers carried out every decade and every year, respectively. The natural population movement, by sex and age groups, will also be considered to know the results in more detail. Likewise, they have been established according to their economic activity, which will link up very well with economic indicators, and thus know the active population by sectors of activity. Another indicator has focused on the linked population, that is, who is not registered but who makes use of the municipal territory for a certain reason. The migrations and migratory patterns of the population are also of interest.

In terms of economic aspects, the change in activity that has taken place in the municipalities of the study area has been analyzed, where the deagrarianization and tertiary economic have been the predominant processes. Attention has been focused on the analysis of the active population by sectors over the years and the evolution followed by each of the sectors.

The urban and real estate evolution in terms of homes and buildings has also been studied, considering various indicators such as the number of homes and buildings built over the years, their age, typology, heights, etc. essential data to know the urbanization processes and detect the change of use of homes. In this case, we have again resorted to the data from the housing statistics provided by the National Institute of Statistics. In addition, the housing and land use data of the Ministry of Public Works have been used, based on the results of the *Corinne Land Cover* projects and the Information System on Land Occupation in Spain and the Cadastre data, accessible electronically.

In addition, images, cartography, and photographs are fundamental instruments in a more qualitative study, since through them it has been possible to compare the past situation with the current one, analyzing the changes and transformations that have taken place, as is the fieldwork to support all the above.

As for the study period, it has been adjusted to the one comprised of the fifties of the last century to the present, since it is broad enough in time to appreciate the most significant changes, but in greater detail in the nearest decades, from 1980 and 1990 onwards. Therefore, a period of more than fifty years will make evident the transformation that has been occurring.

## 4. Consequences of the Urbanization Process

### 4.1. Recent Demographic Dynamics in the Marina Oriental

The current 584,407 inhabitants of Cantabria are very unevenly distributed throughout the territory, with a settlement model that is not balanced. Specifically, our study area currently hosts more than 100,000 inhabitants on a regular resident basis, a figure that rises well above if we consider the population linked to them, as will be discussed later.

As for the population evolution, the set of the seventeen area municipalities of the *Marina Oriental* experienced an important and constant growth until the first decade of the current century. From that moment on, the growth maintained, constant and stable (Figure 3) until we get to the present, which explains the population fixation on this space, slowing down the great growth that until now was taking place.

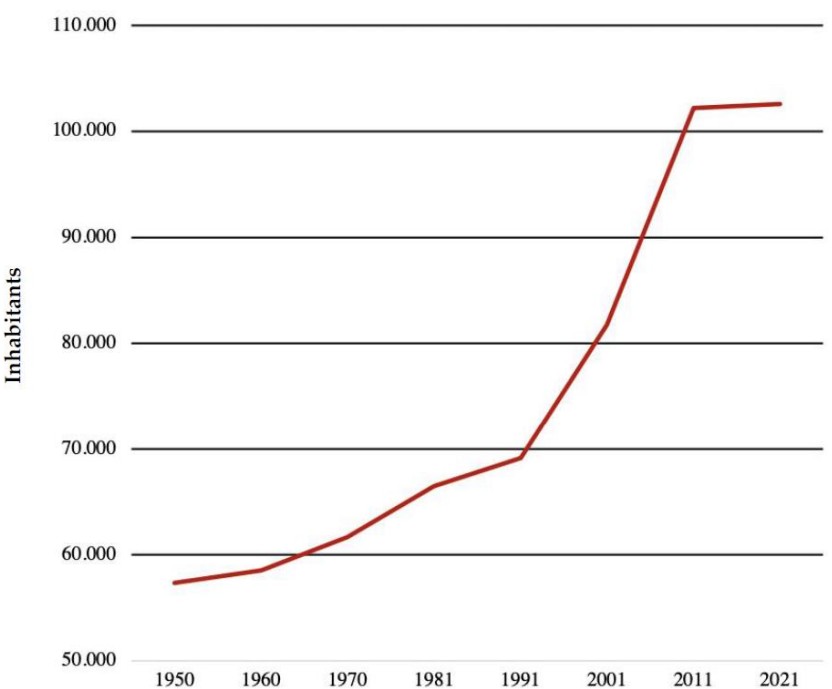

**Figure 3.** of the *Marina Oriental* between 1950 and 2021 (absolute values). Source: own elaboration based on the data of the Population and Housing Census. Series Stories of Population. National Institute of Statistics (INE).

This population variable has great importance within the territory, especially in the municipalities closest to the metropolitan area of Bilbao, such as Castro Urdiales (32,270 inhabitants), Laredo (11,023), and Santoña (11,019), the three cities with the largest population entity within the study area. There are a small number of municipalities, in this case five, Limpias (1974 inhabitants), Bareyo (1950), Argoños (1748), Liendo (1204), and Escalante (762), rural from a strictly quantitative and statistical approach due to their amount of population, which contrasts functionally with the rural municipalities of the interior of Cantabria with a similar demographic volume (Figure 4).

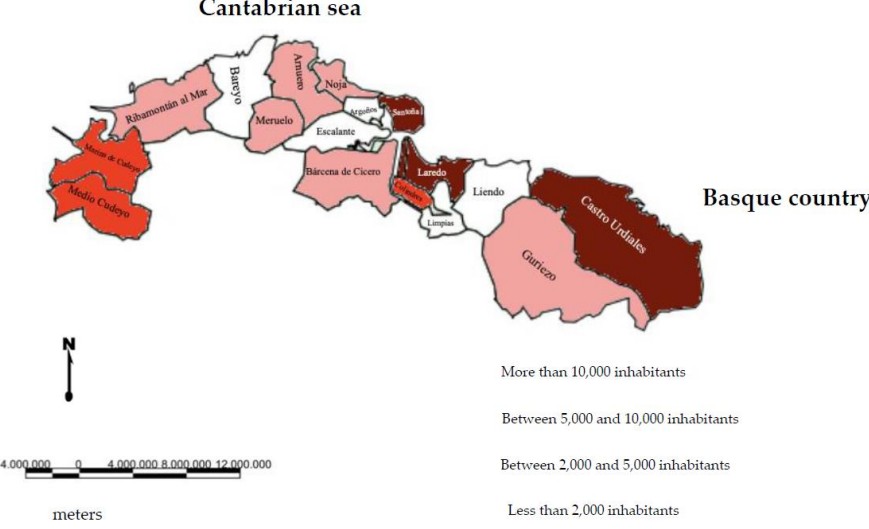

**Figure 4.** Population volume in the *Marina Oriental* de Cantabria, year 2020. Source: own elaboration based on the data and the categories established by the National Institute of Statistics.

It is an area with a great population dynamic, which has experienced, like other important nuclei of the region, high population growth, especially since the nineties of the

twentieth century. A notable example is Castro Urdiales, which maintained stable growth until the nineties, with figures less than 13,000 inhabitants, comparable to Laredo, and, from that moment on experienced a great increase, with highs in 2011 around 32,000 registered inhabitants. However, this city has a real population figure of more than double, counting the population that permanently resides here but has not officially registered its registration, according to the data provided by municipal bodies, such as water and garbage consumption data, which offer an estimate of the real population that the municipality hosts.

The population evolution, of course, has not been followed in the same way by all the municipalities that make up the *Marina Oriental*, since each one has had its internal trend, although they were in general, similar, especially until the end of the twentieth century (Figure 5).

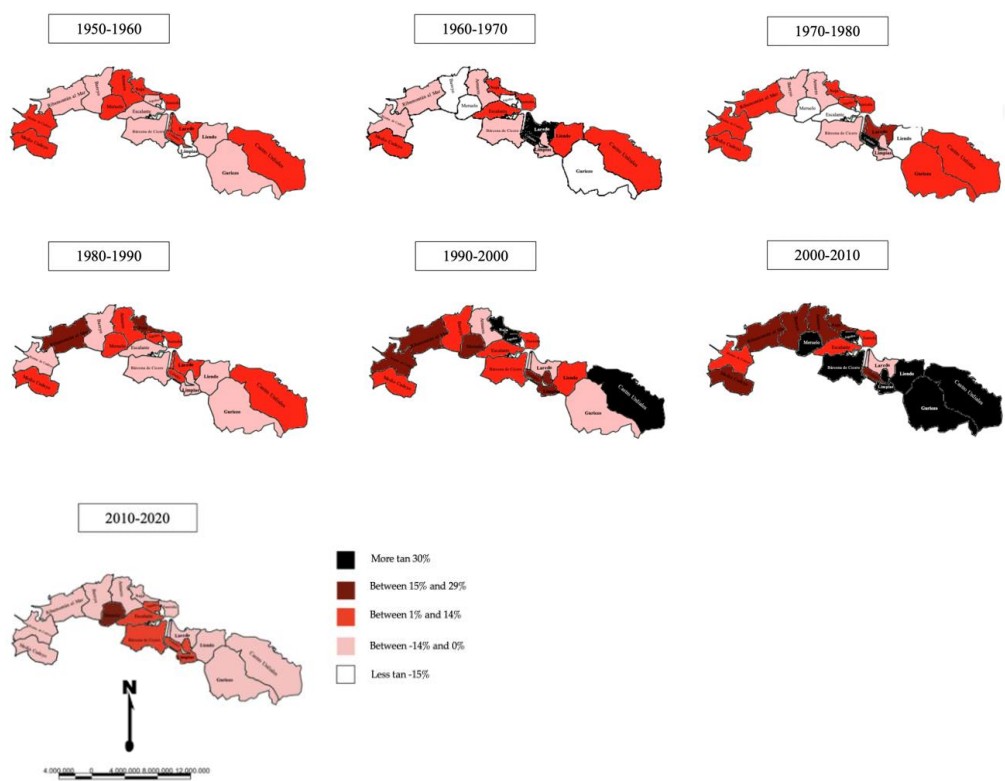

**Figure 5.** Percentage population evolution of the municipalities of the *Marina Oriental* by decades. Source: own elaboration based on the data of the Population and Housing Census. Historical Population Series. National Institute of Statistics (INE).

There are cases, however, in which there are small demographic losses in some of its nuclei, especially in the most urban ones, with population decreases of 1611 inhabitants (−13%) in Laredo and 537 (−5%) in Santoña during the last decade. In these cases, it is not a symptom of decline linked to rurality but of the peri-urban redistribution that has also begun to occur in the small towns of the *Marina Oriental*.

As has already been pointed out before, this area has also been partly nourished by a population of Basque origin, with high population figures, especially during certain times of the year, but it is a floating rather than resident population.

It is essential, therefore, to focus attention on the importance of the linked population[2] in the study area of the *Marina Oriental* of Cantabria, especially because they reside in their second homes for many parts of the year. This circumstance has been evident since the seventies of the twentieth century, but it has intensified since the last decades of the last century, when the second home of a holiday nature had great importance, especially in coastal areas.

Thanks to the improvement of accessibility, transport infrastructures, access to the car, and, in general, the improvement of the standard of living, work movements have increased both in number and in travel distances. So the population is no longer looking for proximity to workplaces but is increasingly opting for places of residence relatively far from urban agglomerations, which are mainly those with the largest job market.

The mobility has increased enormously in Bilbao and its metropolitan area, as well as in the easternmost area of Cantabria, especially Castro Urdiales, place chosen by many Basque popula-tion for their place of residence. Their work destination continues to be mostly the metropolitan area of Bilbao, but they decide to move daily from one place to another, due to its proximity. All this generates great mobility, which is practically impossible to evaluate and account for because this population has not changed the registration of registration [23] and they continue to be registered as Basque inhabitants, even if they live in Cantabria

In addition, the impact of the COVID-19 pandemic that has affected the entire country has altered mobility enormously. Mobility restrictions have been strict and repeated throughout 2020 and 2021, causing major disruptions in people's daily lives and mobility.

In fact, mobility between Cantabria and the Basque Country has increased after the confinement phases established by the State of National Alarm, since, after the opening of the perimeter closures, there was a lot of the Basque population that went to their Cantabrian places of second residence within very specific days, which were those allowed, therefore increasing mobility but also traffic congestion. It is important to point out the duration of this state of alarm, in force in Spain since March 2020, with the impossibility of daily commutes until the central months of that same year.

To corroborate all the above, since the end of 2019, the National Institute of Statistics has considered conducting a mobility study with data obtained through mobile phones. With the outbreak of COVID-19, this study became much more interesting, to know the mobility of people during the declared State of Alarm, obtaining a series of results of notable interest in the eleven areas of mobility within the area of the *Marina Oriental* of Cantabria.

In all areas, the mobility of people who leave the study area increased during the central months of 2020, when mobility was allowed, declining again at the beginning of 2021 and increasing again during the central summer months, when mobility was practically the same as before the pandemic. In terms of volume, the largest percentage of the population that leaves their area of residence daily corresponds to the municipalities of Marina de Cudeyo (35% of its population), Bárcena de Cicero, and Escalante (25%), as well as Colindres (22%); departures are mainly destined for both the metropolitan areas of Santander and Bilbao, especially for work.

*4.2. Changes in Economic Structure and Dynamics*

The most important economic sector in the region today, as has been said, is that of services, with 75.1% of assets in the region in 2020, followed by industrial activity with 14.5%, construction with 7.3%, and, finally, agriculture with 3.1%.

The changes in economic activities that have occurred today have generated the transition from an economy based on the activities of the primary sector, livestock, and fishing, to an economy practically focused on the tertiary sector, on services, especially those linked to tourism.

As for the business structure of the activity between 2009 and 2019, most of the establishments are focused on repair (more than 2500 establish-ments), construction (more than 2000), and hospitality (almost 1500). With the compa-nies, we see a significant percentage of them in 2009 corresponding to construction and trade, both close to 25% of the total companies of the Marina Oriental, followed by the rest in services, with almost 20%. In 2019, the situation varied a little, since the largest percentage correspond to the rest of the services, exceeding 25%, with trade and construction decreasing in importance, the latter below 20% (Figure 6 and Table 1).

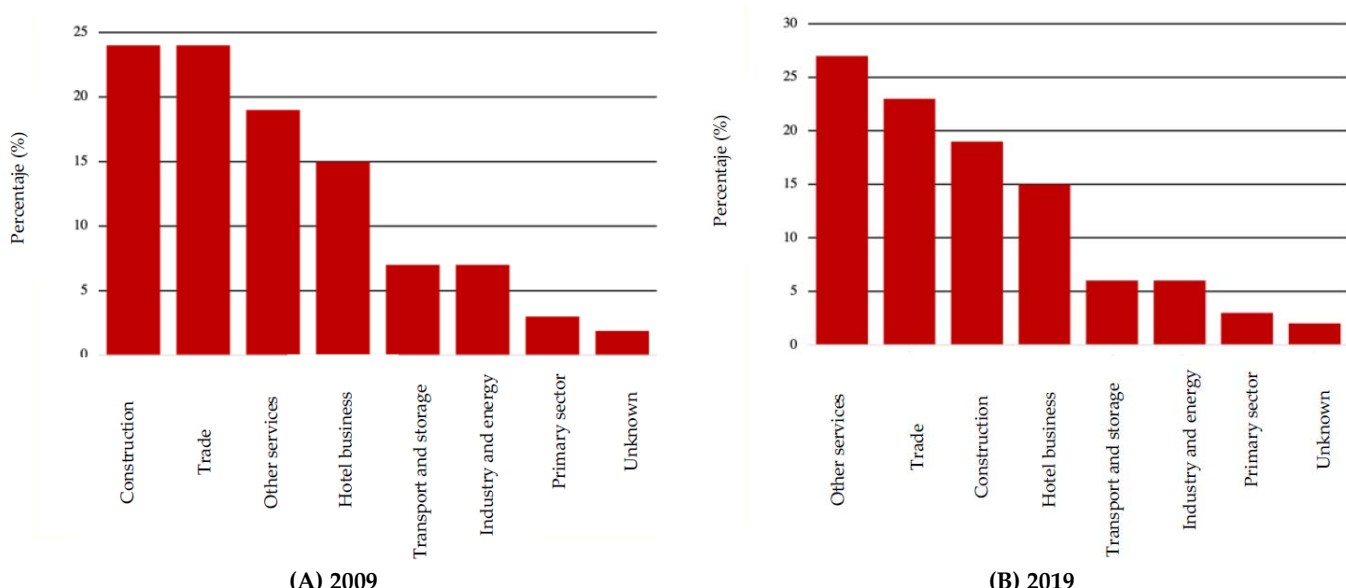

**(A) 2009**  **(B) 2019**

**Figure 6.** Business structure in 2009 and 2019 in the *Marina Oriental* of Cantabria. Source: own elaboration based on the data of the Directory of Companies and Establishments of Cantabria. Cantabrian Institute of Statistics (ICANE).

**Table 1.** Percentage of companies by industrial activity in the municipalities of the *Marina Oriental* de Cantabria in 2009 and 2019.

| | 2009 | | | | 2019 | | | |
|---|---|---|---|---|---|---|---|---|
| | Agriculture and Fisheries | Industry | Construction | Service | Agriculture and Fisheries | Industry | Construction | Service |
| Argoños | 4 | 2 | 2 | 1 | 6 | 3 | 2 | 1 |
| Arnuero | 3 | 1 | 4 | 2 | 4 | 1 | 4 | 3 |
| Bárcena de Cicero | 4 | 5 | 4 | 3 | 7 | 6 | 5 | 3 |
| Bareyo | 5 | 0 | 4 | 2 | 12 | 1 | 3 | 2 |
| Castro-Urdiales | 17 | 19 | 21 | 27 | 8 | 17 | 24 | 27 |
| Colindres | 7 | 7 | 10 | 7 | 3 | 6 | 9 | 7 |
| Escalante | 1 | 1 | 0 | 1 | 2 | 1 | 1 | 1 |
| Guriezo | 1 | 2 | 2 | 2 | 4 | 2 | 2 | 1 |
| Laredo | 10 | 9 | 16 | 16 | 6 | 13 | 15 | 15 |
| Liendo | 1 | 1 | 3 | 1 | 2 | 0 | 2 | 1 |
| Limpias | 0 | 2 | 2 | 1 | 1 | 1 | 1 | 1 |
| Marina de Cudeyo | 11 | 10 | 3 | 4 | 11 | 11 | 4 | 5 |
| Medio Cudeyo | 10 | 14 | 6 | 9 | 9 | 15 | 7 | 9 |
| Meruelo | 3 | 3 | 3 | 2 | 4 | 3 | 3 | 2 |
| Noja | 2 | 1 | 8 | 5 | 1 | 1 | 7 | 6 |
| Ribamontán al Mar | 13 | 4 | 5 | 5 | 15 | 3 | 5 | 5 |
| Santoña | 9 | 17 | 7 | 12 | 7 | 17 | 7 | 12 |
| **MARINA ORIENTAL** | **25** | **17** | **19** | **17** | **20** | **17** | **17** | **16** |

In the study area, as for the whole of the Cantabrian region, the agricultural sector, especially livestock activities, has been of vital importance for the economy. Likewise, fishing is a traditional activity that has been gaining more and more importance weight ver time. The construction of fish canneries has been the fundamental factor in the transformation of the sector, thereby changing the entire fishing system, the production process, techniques, etc., going from a practically subsistence system for the population to being a consolidated and productive economic activity [24,25].

All these changes in the current trend can be understood, in general terms, as a "rural restructuring" [26], referring to a change in the productive structure of rural spaces since the beginning of the seventies, when the processes of deagrarianization in Spain ere beginning to be noticed and, although to a lesser extent in recent times, by increasing mobility between the city and the countryside.

This is evident in the study area through a series of indicators, such as the decrease in the number of agricultural holdings, especially since the nineties, with reductions exceeding 90% between the end of the seventies and the present. If in 1972, we found more than 6000 farms in the study area, what we see today is the presence of just over 1000 farms, that is, a reduction of more than 80% in less than forty years of study, in each one of the municipalities, without exception, especially pronounced since the nineties.

Therefore, there was a loss of importance in terms of the number of livestock farms, which became almost complete in some municipalities, such as for example in Santoña, where there are only twelve agricultural farms left, representing less than 1% of the total farms of the *Marina Oriental*, or Noja or Colindres, with twelve and thirteen farms respectively, also representing 1%.

Another result of the loss in activity comes hand in hand in the variation of the size of the farms is that there has also been a change in the trend, with farms preferably of small size (less than 20 hectares), which constitute 80% of the total farms in the study area. These farms, despite being the most predominant, have also seen the number of medium and even large-sized farms (more than 50 hectares).

Everything mentioned above also translates into a reduction in the useful agricultural area occupied by these farms, which, of course, has been reduced. The useful agricultural area has gone from exceeding 42,000 hectares for the whole of the *Marina Oriental* in 1972 to standing at just over 14,000 by the year 2020 (Figure 7). If it is analyzed at the municipal level, it can be seen how the reduction has been very intense between 1972 and 2020, with municipalities that have practically completely lost their agricultural use area, such as Santoña, Liendo, Guriezo, Noja or Castro Urdiales, with reductions of around 98%.

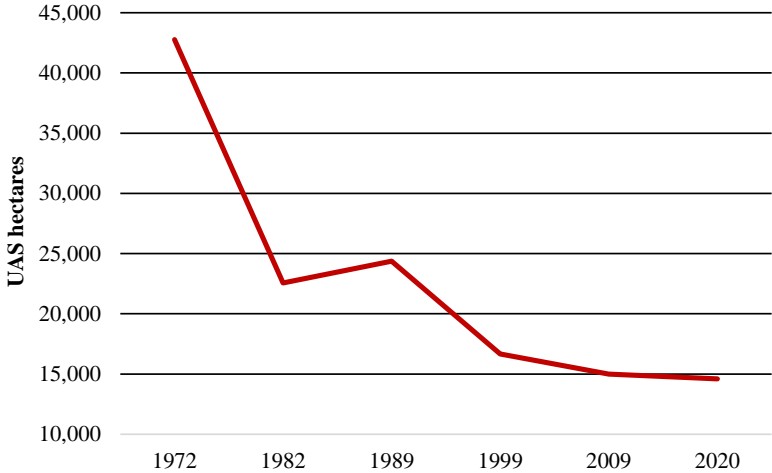

**Figure 7.** Evolution of UAS hectares in the *Marina Oriental*. Source: own elaboration from the agrarian censuses.

In addition, there has been a change in the tenure regime of these farms, in parallel with this process of privatization. Over the years, there has been a change in land ownership towards leasing. As for the age of the holders of the holdings, it is seen that there have also been variations. At the beginning of the seventies in the study area, the highest percentage of owners were of ages between 35 and 54 years; however, at present the trend has turned towards older owners, in this case between 55 and 64 years, which supports that another feature of deagrarianization is the disinterest of young people to continue with agricultural activity [27].

Despite this, the number of cattle has been increasing over the years, from 37,500 in 1982 to 269,640 in 2019, especially highlighting the cattle, the most important within the area of the Marina Oriental, the half of the total and even in some municipalities reaching higher percentages, such as Ribamontán al Mar (73%).

However, dairy quotas have been reduced. These began to be applied after Spain's entry into the European Union, trying to balance supply and demand in terms of dairy production and eliminating production surpluses. There were maximum quotas between 2002 and 2005, above 103,000,000 kilos of milk, to be reduced from there until it stopped being counted in 2015, when was around 96,000,000 kilos, with differences, logically, at the municipal level. Since that moment, milk production data were taken from industry, where the municipalities with the highest contribution are Ribamontán al Mar (26% of the quotas), Marina de Cudeyo (18%), Bareyo (15%), and Arnuero (11%), that is, those with the highest maintenance of the activity.

If the employment structure is taken as a reference, it can be seen how the primary sector has decreased for the entire study area, from 1999 to 2019, with a progressive decrease in its percentage of assets. All this presents nuances, since, despite the generalization in the loss of activity, there have been municipalities that have known how to safeguard the activity, despite not being the one with the greatest weight at present, employing a significant volume of population.

Internal differences can be seen within the study area. On the one hand, municipalities that from 1999 to 2019 have greatly decreased the number of employees in the sector, such as Argoños (from 6% to 4%), Colindres (from 14% to 6%), Guriezo (from 7% to 1%), Laredo (from 11% to 3%), or Santoña (from 16% to 6%), taking into account that in Colindres, Laredo, and Santoña, many of the employees in the primary sector are not agricultural but fishing. On the other hand, the westernmost municipalities increased the numbers of agricultural employees between these dates, such as Arnuero (from 0% to 3%), Bárcena de Cicero (from 2% to 4%), Escalante (from 2% to 4%), or Ribamontán al Mar (from 0% to 5%).

The highest value corresponds to the municipality of Bareyo, which has the highest percentage of agricultural active population with 6.7% (Figure 8). On the contrary, other municipalities present percentage figures in some worrying cases of the abandonment of the activity, as for example in Noja, with percentages of agricultural population of 0.2%, Castro Urdiales with 0.3%, or Limpias with 0.6%, insignificant percentages for an activity that has been the economic pillar for this region in not-so-distant times. Faced with this fact, little by little, alternatives are being created to generate work and activity in rural and peri-urban spaces, so that they do not completely lose the essence and rural identity that has characterized them for many centuries (Figure 6).

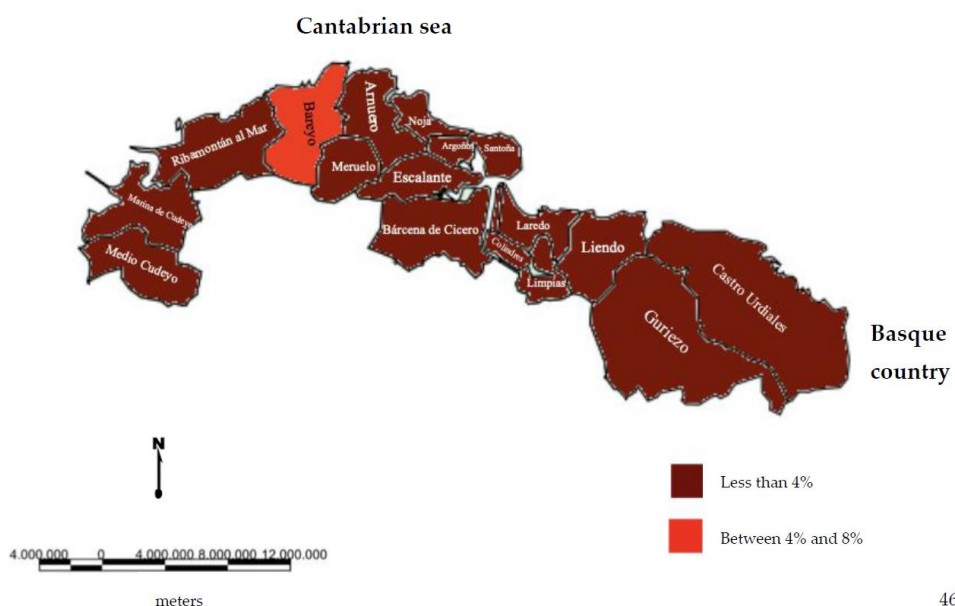

**Figure 8.** Percentage of agrarian population in 2020 in the municipalities of the *Marina Oriental* de Cantabria. Source: own elaboration based on the exploitation of microdata of the active population surveys. National Institute of Statistics.

*4.3. Tourism Tertiarization*

Deagrarianization and the loss of agricultural activity are being counteracted with a boom and importance of tertiary and service activities, that is, with a socio-economic outsourcing, especially focused on tourist and hotel accommodations. The territory of the *Marina Oriental* hosts 40% of the total hotel places that Cantabria has and almost 12% of the places for rural tourism houses. Of this 40% of hotel places, two municipalities stand out with more than 20% of places, Arnuero with 20.6% and Noja with 25.6%. They are areas with many places due to the development of hotel accommodations that have been built in both municipalities and that offer tourists an alternative of accommodations outside of residential tourism that prevails in the rest of the territory.

Another of the most successful accommodation models within the territory of the *Marina Oriental*, where there is a large number, the campsites. This accommodation model is a relatively recent tourist variety in Spain, since it has been developed over the last decades, especially in the areas closest to the coast, becoming a fully consolidated model. Specifically, within the territory of the *Marina Oriental*, we find 23 campsites, that is, 46% of the total.

This service sector, which already had the highest percentages of assets at the end of the nineties (about 60% in 1999), saw a slight reduction for the year 2009, increasing later in the year 2019, occupying practically 80% of the assets of the *Marina Oriental* of Cantabria. The almost total percentage focuses on the demand around the services sector, with percentages above 60% in all cases.

In addition, it is appreciated that the unemployment figures are focused especially around the sector with the largest number of employees, the services sector, higher than 80% on certain occasions, as in the case of Argoños (81%), Bareyo (82%), Meruelo (83%), Noja (86%), or Ribamontán al Mar (81%), which generates a great problem linked to the loss in the diversity of economic activities and also high unemployment figures. To this is added its seasonality, since it is practically developed at certain times and periods a year, and the rest of the time, many of them, especially in the most touristic municipalities of the *Marina Oriental*, struggle to maintain themselves and be viable.

Regarding the structure of establishments and companies, the service sector constitutes the almost complete predominance of the study area, which reached very high percentages in 2009, such as in Laredo, Medio Cudeyo, or Santoña, where establishments of this type

exceeded 70%. It also highlights the percentage of establishments dedicated to construction, at a time when the crisis in the sector had begun, but the companies related to it were still very numerous. In 2019, the situation remains similar and services again have the highest percentages of importance in terms of establishments, all of them with percentages higher than the 48%.

As for the industrial sector, it has experienced a generalized decline during the years of study within the area of the *Marina Oriental*. Among them, the area of the mouth of the Asón stands out, such as the installation of SEG Automotive Spain in Bárcena de Cicero, as well as various industrial estates such as in Medio Cudeyo or Castro Urdiales. Despite this, the predominance in terms of industry is formed by small facilities. While assets increased from 1999 to 2009, especially in the municipality of Bárcena de Cicero (with more than 35% of employees) or Colindres (with almost 35%), they decreased in 2019 in general, with the lowest weight of employees in the sector being the municipality of Noja (with 4%) and the one with the most Santoña (with almost 20%). The percentage of assets in this sector increased until the impact of the financial crisis in 2009, and from then, until the present, it was reduced again.

In order of importance, industrial activity is followed by construction, which, although it has decreased since 2009, due to the crisis in the sector, still has a lot of weight within the area.

Construction, like the industry, saw its assets increase until 2009, again to be reduced today, below the values it had at the beginning of the analysis. The construction sector saw its figures increase in 2009 to later decrease in 2019, its highest in the municipality of Marina de Cudeyo (more than 15%), with the rest of the municipalities being around 5% employed.

For its part, the services sector, which already had the highest percentages of assets in 1999 (about 60%) had a slight reduction around 2009, increasing later in 2019, occupying practically 80% of the assets of the *Marina Oriental* of Cantabria, which endorses the importance of this sector within the study area, with almost absolute weight over the rest of the sectors. In addition to being the one with the greatest importance and weight in all the municipalities of the study area, it is the sector with the greatest increase for the analyzed period, with continuous promotions, in all the municipalities, with more than 65% of employees around the sector.

## 5. The Intensification of Urbanization Processes

As has already been reiterated, most of the changes that have been exposed have had their origin in the metropolitan growth of Bilbao (Figure 8) and in the construction of infrastructure that connected both spaces in a fast way since its construction, the Cantabrian Highway, A-8, which increase the strong urbanization process that occurred in this space in recent decades.

According to the study of the National Institute of Statistics (INE) on functional urban areas[3], Bilbao covers a large part of the territories that make up the *Marina Oriental* of Cantabria, as we will see below (Figure 9). The metropolitan area of Bilbao is an urban space that covers an area of more than 300 square kilometers, made up of about thirty municipalities in more than one million people live. Within this space, there is the presence of municipalities belonging to the *Marina Oriental*, such as Castro Urdiales and Guriezo, among others, which are today functionally integrated completely into the metropolitan area of Bilbao [18]. It is metropolitan growth that has exceeded administrative limits and has grown enormously, exceeding previous expectations and esti-mates.

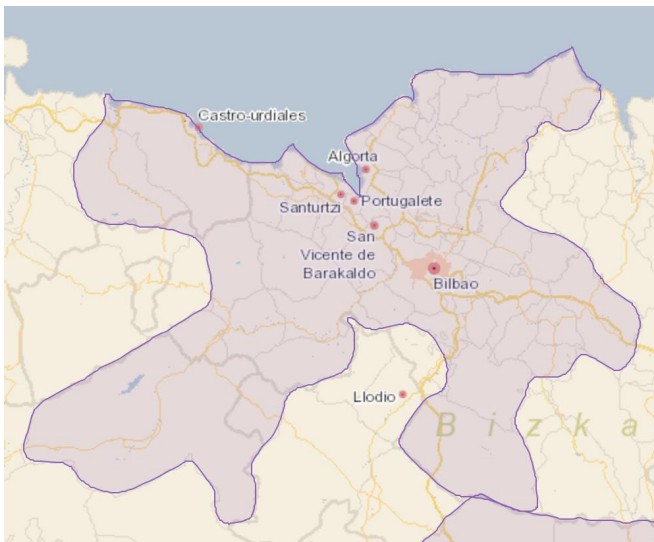

**Figure 9.** Delimitation of the metropolitan area of Bilbao (Functional urban area). Source: National Institute of Statistics.

In its urbanization dynamics, the role of secondary housing or into this space, with values that reach such high figures in certain municipalities as to consider them "paradigmatic examples".

In the *Marina Oriental*, as happens in many other places on the Spanish coast, such as the Mediterranean, Levantine, and Andalusian case, the intensification of recent urbanization processes is linked to tourist activity, but they also have multiple elements linked to holiday residentialism. Many municipalities in this territory receive a large population, especially of Basque and Madrid origin, who have opted for the acquisition of homes for the enjoyment of leisure time in a coastal destination like this or as an investment [28,29].

As can be seen in the graph (Figure 10), the main growth in the housing stock in Cantabria and in the *Marina Oriental* has occurred during the last decades in proportions higher than the increase in the population.

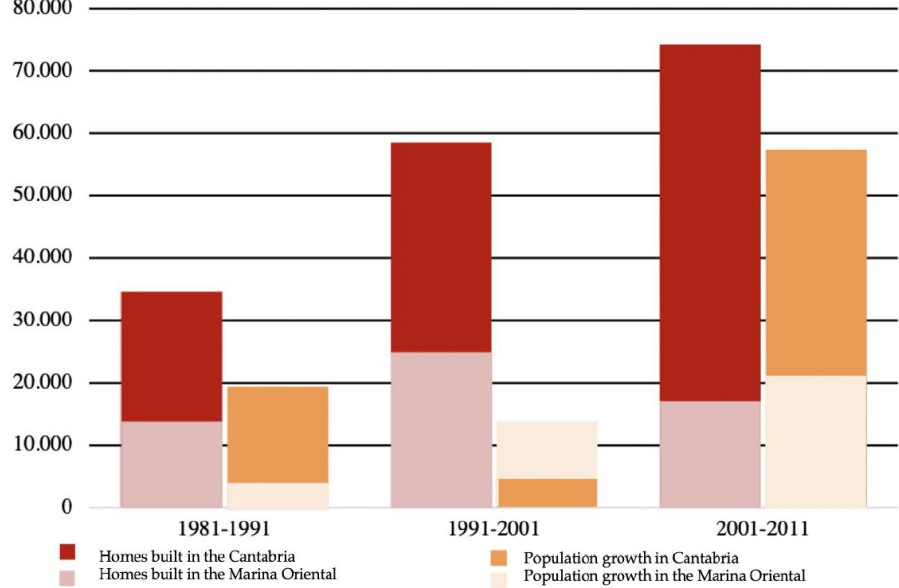

**Figure 10.** Evolution by decades of the total number of homes built and population growth in number of inhabitants for Cantabria and the *Marina Oriental.* Source: own elaboration based on the data of the National Institute of Statistics.

As can be seen, the number of homes built in the region since the eighties of the twentieth century has always been higher than population growth, with great differences between both variables, especially between 1991 and 2001, when the volume of homes built grew greatly and the population barely did. Housing growth continued in the decade between 2001 and 2011, reaching its peak for that period, with the construction of more than 70,000 homes.

In the *Marina Oriental*, the number of homes built was higher than the population growth between 1981 and 2001, reaching its maximum in the decade between 1991 and 2001, with a construction of more than 25,000 homes.

In the Marina Oriental is the development of young residential park, since most of the houses have been built since the ninenties, with an average, therefore, of about thirty years for the total age of the houses. It was during the nineties, when most homes were built (25%), followed in importance by the first decade of the two thousand, with more than 20% of homes. In fact, it highlights the existence of years in which the percentage of increase in new homes with respect to the total real estate stock has been higher around the *Marina Oriental* than in Cantabria, especially since the eighties (Figure 11).

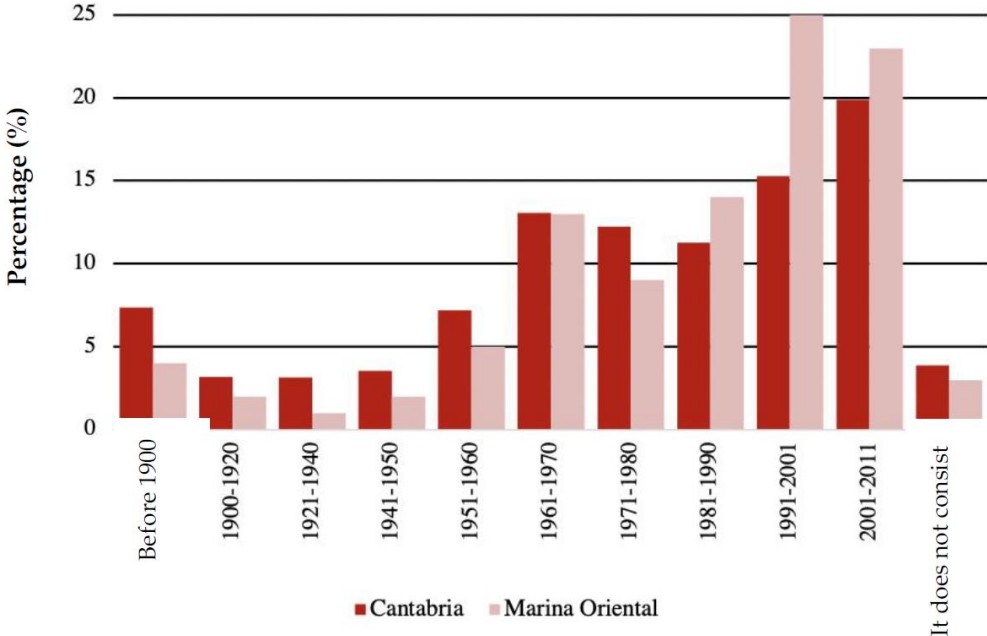

**Figure 11.** Comparison of the evolution by decades of the total number of homes built in Cantabria and the *Marina Oriental*. Source: own elaboration based on the data of the National Institute of Statistics (INE).

The great dynamism of residential construction that took place in the study area, especially since the nineties, resulting in the fact that practically all their residential park has been built since that moment, as in the case of Noja (Figure 11) or Castro Urdiales. For its part, Laredo is a previous tourist residential park, from the sixties and seventies, located around Salve Beach (Figures 12 and 13).

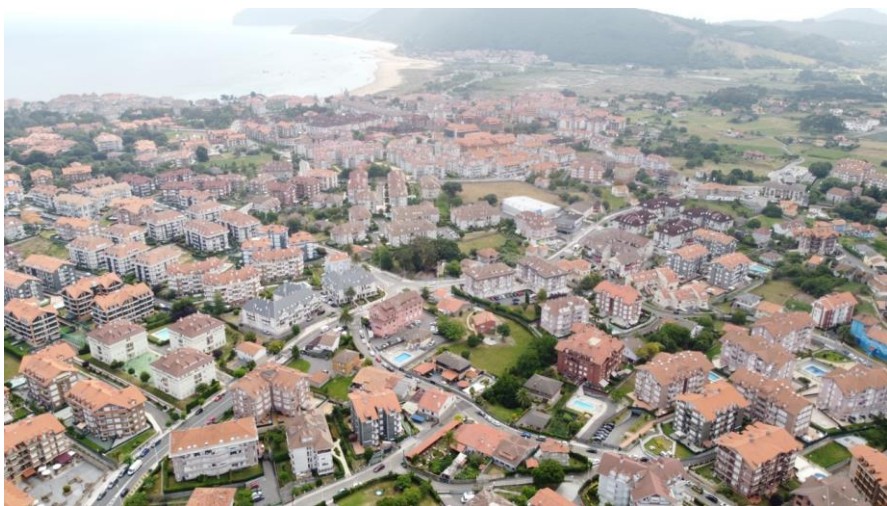

**Figure 12.** Residential construction in Noja. Source: own images.

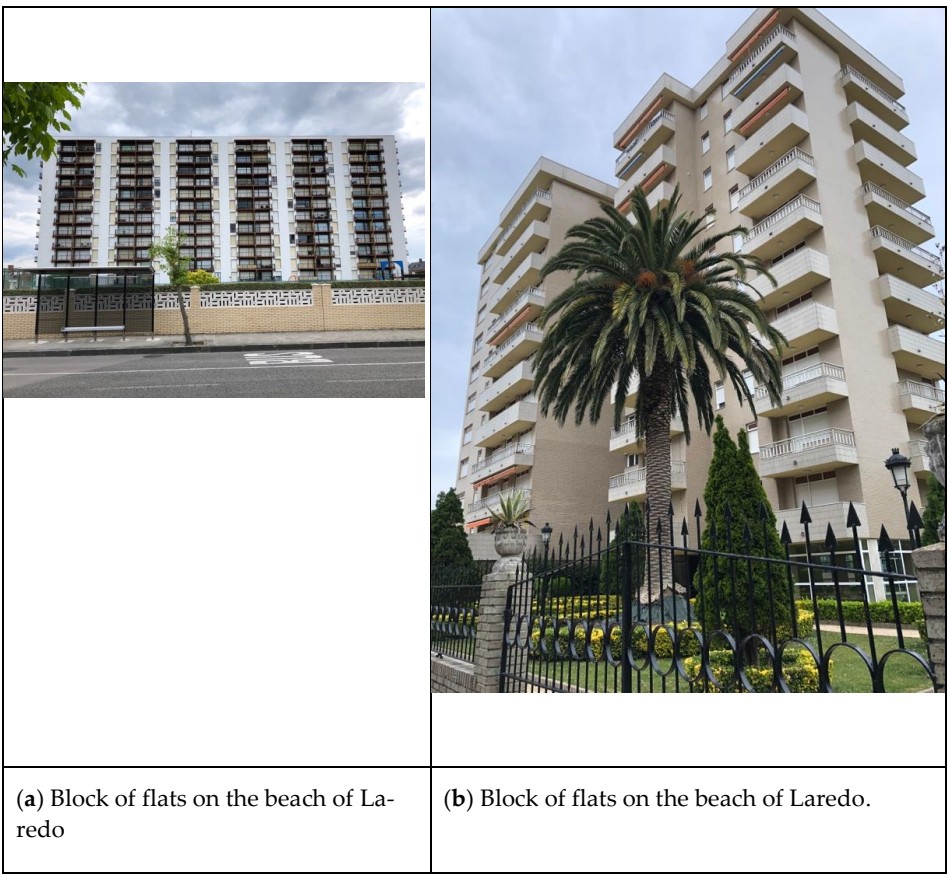

| (**a**) Block of flats on the beach of Laredo | (**b**) Block of flats on the beach of Laredo. |
|---|---|

**Figure 13.** Residential construction around Salve Beach (Laredo). Source: own images.

The Multi-family type housing (50% of the total housing stock) it has a rural profile, since they have been established in municipalities that, in most cases, do not exceed 5000 inhabitants registered. Since the eighties this model has been consolidating to take advantage the most of the available buildable land. There is also a large presence of isolated housing, usually one or two floors, with such as enclosures for stables, garages, or storage areas.

This process has caused great transformations in the territory and the landscape of the Marina Oriental, especially on the rustic land, which has sometimes been converted into

urbanizable and urban land, called "rururbanization"[4] processes [30], the initial mechanism of diffusion and dispersion of the city in the rural space [31].

This process has manifested itself in a very intense way in Marina Oriental gener-ated, above all, by the lower price of buildable land in relation to in the Basque Country, as well as by the unique socio-political circumstances of the Basque Country in previous decades. This entails some problems derived from the superiority of floating linked population that consumes services and equipment but does not contribute with its taxes to construction and maintenance.

At this point it is necessary to talk about the importance that second homes have acquired in the *Marina Oriental* of Cantabria. As mentioned before, the volume of secondary housing throughout Cantabria, but especially in the *Marina Oriental*, is highly substantial, exceeding on many occasions the number of main homes and constituting in some municipalities practically all the homes in its real estate, as in, for example, Laredo, where secondary homes represent between 60 and 80% of the total, or Noja, with 91% of secondary homes in 2011 (Figure 14).

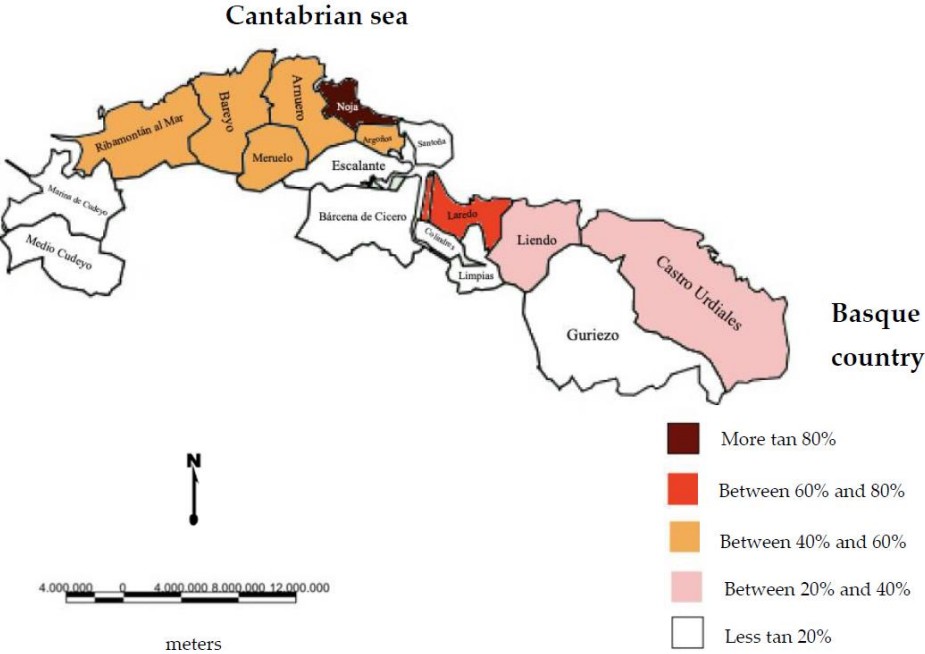

**Figure 14.** Percentage of secondary housing in the municipalities of the *Marina Oriental* for the year 2011. Source: own elaboration based on the population and housing censuses. National Institute of Statistics (INE).

In fact, the municipality of Noja occupies number one in the ranking of the entire Spanish territory as the municipality with the highest percentage of secondary housing in relation to the volume of the housing stock, above highly touristic municipalities such as Salou in Tarragona, Benicasim in Castellón, Mojácar in Almería, or Punta Umbria in Huelva, among others, for the year 2011.

As for the use given to these secondary homes in the *Marina Oriental*, it is characterized by being seasonal. They are occupied during the summer period, at other holiday times and even on weekends, although there are exceptions, such as Castro Urdiales where many of the homes registered as secondary are, in fact, permanent homes of the non-registered population.

In residential construction stress the multifamily block housing with between three and five floor, except in Laredo, which is sometimes described as the "Benidorm of the north", where ten levels are frequently exceeded. With this typology, the builders were able to make more intensive use of the soil than with the construction of isolated single-family houses, which consume a large amount of land.

The little controlled form of urbanism that has taken place in the *Marina Oriental* has had great effects on the landscape, creating totally artificial scenarios, the result of real estate speculation, whose objective is the maximum use of the land and not the sustainable management of it, hence the predominance of the most intensive architectural typologies. It has not been uncommon for this urbanization process to sometimes produce irregularities that have led to the demolition of homes classified as illegal, as well as the presence of many works stopped. All this has occurred despite the validity of several legal norms of urban and territorial planning in the Coastal Management Plan (POL), which was approved in 2004 and which can be described, without fear of equivocation, as the most important instrument in terms of management and planning in the *Marina Oriental*. The fundamental problem derives from the fact that the intense and accelerated process of urban expansion has developed in poorly adequate conditions, especially using the normative figures of territorial planning at the municipal level, some of them obsolete or poorly adapted to the great urbanizing pressure: only 47% of the municipalities of the *Marina Oriental* have a General Urban Planning Plan (PGOU), while 53% only have Subsidiary Planning Rules (NNSS).

The articulation of the territory in the study area of the *Marina Oriental* follows certain patterns and trends that it is necessary to analyze the concept of "territoriality", a complex and important concept, which allows greater flexibility when cataloguing various territories in relation to different criteria. Thus, within the study area, considering this territoriality and the characteristics that, on the ground and in fieldwork, are verified; the territory of the *Marina Oriental* will be articulated in three large groups or zones (Figure 15).

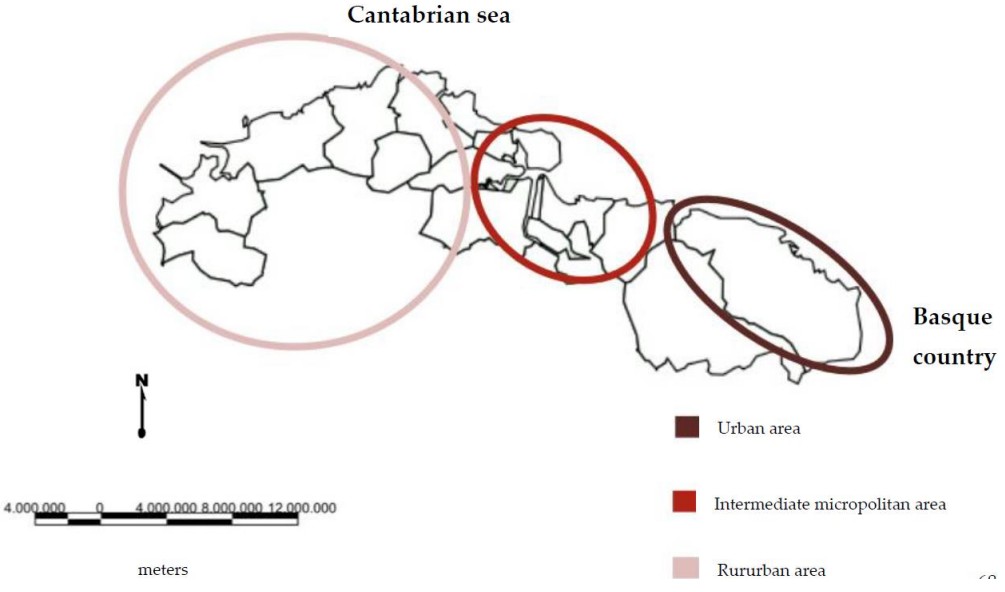

**Figure 15.** Areas/sectors of the *Marina Oriental* based on its territorial articulation. Source: own elaboration based on the cartography of the government of Cantabria.

On the one hand, the evidently more urban area is to the east, with the urban nucleus of Castro Urdiales; on the other hand, the conurbation formed in the intermediate part by the nuclei of Santoña, Laredo, and Colindres, and the area further west, eminently more rural, made up of the territory of the old historical region of Trasmiera.

## 6. Discussion

The discussion leads us, in short, to see how the hypothesis that has been proposed is verified and at the same time has coherence with the bibliographic framework.

Firstly, according to the methodological framework, the objective was the geographical analysis of a coastal territory of Cantabria, articulated by seventeen municipalities of diverse nature and typology but which act as a common space, well integrated and

differentiated from the rest of the Cantabrian territory, positioning itself as the focus of greater development in recent decades within the region, with great attraction, especially in the metropolitan area of Bilbao and to a lesser extent in the metropolitan area of Santander.

The sea has played an important role here, hence the designation as *Marina*, a term in recent use, understood as that territory that has proximity to the sea.

All this will have a direct and important impact on the population, a variable of great importance in the development of this study, the result of the intensification of urbanization processes in this space, being the area of greatest growth and population dynamics in recent decades. In this way and recently, especially since the middle of the last century but more intensely since the nineties, this space of the *Marina Oriental* of Cantabria has experienced great population growth in each one of the seventeen municipalities of study.

There has been remarkable population growth in the study area, earlier in some municipalities than in others [32]. Firstly, in municipalities such as Noja, which has gone from 1300 inhabitants in 1980 to 2100 in 2001, Laredo or Castro Urdiales (from 13,050 inhabitants in 1980 to 21,081 in 2001), in the last years of the twentieth and in the early twenty-first century. Then, after the first years of the XXI century, in municipalities such as Colindres (from 6900 inhabitants in 2001 to 8100 in 2011), Meruelo, and Bárcena de Cicero (from 2400 inhabitants in 2001 to 4100 in 2011), which, after the previous growth of the nuclei mentioned in the first place and being located very close to them, tend to grow later.

It is then that the *Marina Oriental* took off in importance, thanks to its strategic location, with connections to both the Basque Country and Asturias, greatly differentiating itself from the rest of the region, exploiting and transforming the soil, acquiring great importance to the linked population, especially of Basque origin, a link that has a lot to do with the secondary housing that this population has in the environment of the *Marina Oriental* of Cantabria.

All this has brought about changes in economic development, moving from an economy based on the primary sector to a tertiary and very seasonal, converting its activity to adapt to needs and new demands. The most important change occurred in the space of great tradition and rural heritage, which was dedicated almost exclusively in the past to the agricultural sector; since the seventies, when the territory was rearticulated because of the pressure of new demands and activities that generate great transformations on the territory, the territory took advantage of the productive value they possess but with new uses.

This led to a great increase in real estate in the *Marina Oriental*, with housing growth above that of population. This expansion has been generated mainly thanks to the metropolitan growth of Bilbao, extending beyond the borders of the Basque Autonomous Community itself.

All this has generated a complex urbanization process, developed in stages over time, triggering an intensification where settlement has taken place closer and closer to the coast, in a highly valued space, turning the study area into a tourist destination for a large number of people, favored by the large residential value it has and generating with it, of course, important impacts, from socio-demographic to economic, technological, and environmental, among others.

The dynamics of the *Marina Oriental* has grown above the growth of the population itself, under a model in which diffusion has prevailed over coastal spaces, with great intensity and with direct consequences on landscapes, territory, and economic activities. This has resulted in a growth in residences, which has intense especially since the nineties of the last century, favored by the good communications and the best price of the housing that we find in Cantabria in relation to that of the Basque Country, leading to a constructive furor, which has not respected, sometimes, the spaces that must be protected.

We cannot forget the importance of secondary housing in this area of the *Marina Oriental*, with percentages sometimes higher than main houses. But all this boom in secondary housing entails a series of impacts and repercussions, such as the lack of endowments and services, especially during the summer season, direct and irreversible consequences on the

landscape, due to the ex-novo construction that has been developed, exceeding the capacity of reception many times and prioritizing quantity over quality, to welcome mass tourism. However, not all aspects are negative, and all this has slowed down the rural exodus that was developing in many of these coastal rural areas, increasing jobs, especially focused on trade and services, and increasing mobility and endowments.

Therefore, it has been demonstrated that this space has a territorial system forged and formed over centuries, undergoing great transformations, including social, economic, urban, and landscape transformations, especially since the middle of the last century but more intensely and deeply since the seventies of the last century. These transformations occurred in a geographical space inherited from past times, which has been modeled according to the demand that has been developing, clearly modifying its character [33,34].

Therefore, a productive redefinition confirms the importance of this space within the whole of the region [35], one of the most changed since the middle of the last century, the fruit of the intensification of urban processes, as a result of the metropolitan growth of both the region and Bilbao, in this case the one with the greatest impact on the area, driven by, to a great extent, in addition to the improvement in living conditions and the attractiveness that the Cantabrian coast possesses, the improvement in communications, in this case thanks to the construction of the Cantabrian highway, with the consequent generalization of the automobile and the lower price of Cantabrian land compared to Basque land.

## 7. Conclusions

As has been demonstrated throughout this article, the research has been carried out on a space on which there were hardly any previous studies but which, nevertheless, presents a great strength, being considered of great importance and relevance and being extrapolable to other spaces and territories due to the importance it possesses. This is presented as a weakness, due to the scarcity of the bibliography and the need for a more complex research work.

It is a space, which, as we have seen, has undergone a great transformation in a few decades, totally changing its image and its activity, adapting to the new residential and tourist uses that have been demanded.

A series of highly relevant findings have been discovered, highlighted, and explained throughout this article, especially within the urban framework, with cases that can be considered models within Spanish territory, with great importance of secondary housing and the related population.

It is also important to reflect on the fact that this territory must be protected and cared for, despite having experienced urban processes that are already irreversible within a totally coastal space, which it is necessary to value and protect, avoiding seasonality and encouraging again the sectors that have lost importance over time. In this case, the planning has not been specifically addressed, but it would remain open for possible articles in the future, since this is a truly interesting topic.

In addition, it is necessary to cite a series of limitations and inconveniences that we find in research of this type, especially focused on the absence of certain quantitative data in official sources, which have been solved by resorting to other sources of a local nature or companies linked to certain activities (water consumption, garbage collection...), to verify that everything developed is real and truthful.

Therefore, it is important to demonstrate the importance of this space of the *Marina Oriental*, concluding that all the processes that have been developed fit and correspond to the dynamics of the economies of the most modern countries, comparing the situation of this space with other countries both in Europe and other continents.

**Funding:** This research received no external funding.

**Data Availability Statement:** Statistical data of various nature, as indicated in the article, have been obtained from the National Institute of Statistics (https://www.ine.es; accessed on 21 October 2022). Data have also been extracted from the National Geological Institute (https://www.ign.es/web/ign/portal; accessed on 21 December 2022). and the Cantabrian government, specifically the land legislation (https://mapas.cantabria.es; accessed on 21 October 2022). Data have also been extracted from the Cadastre, especially for real estate analysis (https://www.sedecatastro.gob.es; accessed on 21 October 2022), as well as the Land Occupation Information System (http://www.siose.es/presentacion; accessed on 21 October 2022), and the Corine Land Cover project (https://land.copernicus.eu/pan-european/corine-land-cover; accessed on 21 October 2022), and field work is also important.

**Conflicts of Interest:** The author declares no conflict of interest.

## Notes

[1] The new and multiple faces of the rural cannot be seen as a finished work. A new vision of the rural is underway, which proposes a new conception of productive activities, especially those related to agriculture, and an equally new perception of the "rural" as a heritage to be enjoyed and preserved [21].

[2] According to the National Institute of Statistics, the linked population of a municipality is defined as "the set of people who have connection with it either because they reside there, because they work or study there or because they usually spend certain periods of time there (holidays, weekends...) during the year.

[3] The Functional Urban Area (AUF), formerly known as the Large Urban Zone (LUZ), consists of a city and the municipalities that form its functional environment, specifically of work influence. The objective is to have an area with a significant part of the resident employed population that moves to work in the city under study.

[4] Occupation of rural areas by a population that has links with the city for mainly work reasons and that, thanks to the improvement in the means of communication and transport infrastructures, moves daily to it without having to change their residence.

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
