# Peer review of "Urban Rural Interaction: Processes and Changes in the Marina Oriental of Cantabria (Spain)"

_land, doi:10.3390/land12010166_

Round 1

Reviewer 1 Report

The text deal with a case study  of coastal urbanisation and territorial and socioeconomic transformation of great interest, centred on the Spanish Cantabrian coast. Most of the studies of tourist urbanisation  in Spain refer to the Mediterranean coast, the Canary Islands and Balearic Islands, but there are few that analyse  the situation in the Atlantic coast. From this point  of view ,the issue is relevant . The so-called geographical method is applied to a supra-municipal reality, with abundant statistical information and graphic representations, wich allows for a solid empirical analysis.

However, in order to strengthen the scientific character of the text and the scope of its contribution beyond the local study, the paper should introduce  the followind improvements:

- The inclusion of a conceptual-theoretical section whith the definitión of the differentiation between rural and urban areas, an issue that is now inadequately coverd in section 4. Results.

- Systematically and clearly set out the research methodology and a table with a detailed account of the sources used their orogi and reliability

- Contextualise the case of the Cantabrian coast in similar processes on a European and global scale, thus strengthening the bibliographical references in the text.

- There are important statements in the conclusions that are not addressed and demostrated in the results, namely that this is a territory forged and shaped over enturies.

Author Response

Thank you very much for your helpful comments and suggestions. I really appreciate the time and effort you have put into helping me improve the quality of the paper. I have paid close attention to these observations and have implemented your suggestions as far as possible.

It has been a great help to me and I appreciate your effort and work in indicating point by point everything that I should improve

I have followed your improvement scheme, trying to make the article better.

Next, I will respond to your comments:

In the introduction, I considered talking about the urban process about the environment and animals, as you indicated, realizing the importance and the relationship that this implies.

In addition, I have divided the discussion and conclusions, since, as you said, the article becomes clearer when I explain it separately and not all unified.

I have also improved the general structure, introducing aspects that he pointed out to me, such as the representation of figures.

I hope that with all this, there has been a valid article for you.

You could find the response in the attachment.

Reviewer 2 Report

The manuscript is well written and enough well structured, anyway some revisions have to be done before going ahead.

Firestly, I suggest you to consider to talk about also in the introduction section how urban process may affect on only human but also environment, human, and animals according to a One health approach just few words. Here it is a couple of work that I suggest you to include in the introduction performing an important state of art.

- https://doi.org/10.3390/ani12081049

- 10.1109/mcg.2004.1255801

Avoid spaces in the abstract

Then, I strongly suggest you to divide discussion and conclusion in two separate sections.

Please consider to include reference system, datum, representation scale and nominal scale per each map reported.

Author Response

Thank you very much for your helpful comments and suggestions. I really appreciate the time and effort you have put into helping me improve the quality of the paper. I have paid close attention to these observations and have implemented your suggestions as far as possible.

It has been a great help to me and I appreciate your effort and work in indicating point by point everything that I should improve

I have totally followed your improvement scheme, trying to make the article better.

I have contextualized a broader theoretical framework, realizing that in this way I attract a wider audience. In addition, I have made the objective, as well as the central hypothesis of the article, clearer. I have also expanded the bibliography, realizing that it was very scarce in the face of an investigation like this.

As I have indicated, I have completely followed your suggestions for improvement, trying to make the article valid, realizing that all of them have been productive and effective in the face of an improvement.

Thanks again for your help and effort with me.

You could find the response in the attachment.

Reviewer 3 Report

The aim of this paper is to analyze the intense urbanization experienced in a dynamic and well-communicated rural area of the Spanish Cantabrian coast. The topic is interesting and, as a case study, the delimitation of the time frame and the spatial scope are appropriate. Likewise, the statistical sources used have allowed us to obtain significant and relevant results.

In its current version, the article has little capacity to attract the attention of an international audience, given that it is not contextualized in a broad theoretical framework. It lacks the statement of a clear objective and does not establish starting hypotheses to discuss and attempt to verify in due course. This sometimes leads to a predominance of description over interpretation. The bibliographical references are insufficient (thirteen), of which only two have been published in the last five years and only one of them in an international journal of high scientific impact. A good way to solve these deficiencies is to carry out a bibliographic search oriented by the key words of the article in international databases such as Scopus or SJR.

The following is a list of aspects that should be reviewed in detail:

1. It is advisable to reduce and restructure the abstract, clearly indicating the justification and objectives of the research, the methodology used, the main results and the conclusion.

2. In the introduction it is necessary to ask one or more research questions. These questions or starting hypotheses should be made in the light of the consolidated and most relevant theoretical-conceptual contributions in the national and international literature.

It is also possible to include the delimitation and characterization of the field of study in this introductory section. Finally, the objective of the work should be clearly established.

3. It is surprising that there is no bibliographical reference to the study area (section 2), in relation to its geographical characteristics.

The map in Figure 1 has no scale or geographic orientation, nor is the source indicated. The same is true for figure 2.

4. With respect to section 3 (material and methods), it is necessary to explain, since it is considered essential for obtaining results, what the fieldwork consisted of.

The sentence that begins on line 147, dedicated to economic aspects, is however reduced to issues related to education. The sentence is a bit confusing.

In line 160, the use of images and photographs is mentioned; however, it is not specified in what way they have helped the research, nor have they been used to illustrate some aspects of the results.

5. In section 4 (Results), the text from line 168 to 193 is theory, not results; it is better to use these reflections and references to frame the research questions suggested for the introductory section.

6. The commentary on Figure 3 states that the field of study has experienced steady growth until the first decade of the 21st century, but does not explain why it has stagnated since then.

7. In the title of Figure 3 the term "Eastern Navy" appears, while in other parts of the text the Spanish expression is used; this nomenclature needs to be standardized.

8. In lines 219 and 220 a statement is made that is not supported by bibliographic or statistical reference.

9. In line 227 and subsequent lines, aspects that should be dealt with in the discussion section are advanced.

10. Figure 5 lacks a source.

11. In lines 259 to 263 there are very interesting theoretical considerations, although they do not have bibliographical references. It would be convenient to move these reflections to an introductory section, as suggested above.

12. In line 277, in relation to the State of Alarm, it would be convenient to explain briefly what it consisted of and during what periods it was active, since the dates are important for the discussion, in the sense of the migratory flows detected during certain periods of 2020.

13. In section 4.2. it would be convenient to introduce tables where the figures that are now described literally are collected, in order to make a general interpretation of them.

14. The paragraph from line 312 to 317 needs to be supported with some quality reference.

15. I do not understand what is said in line 321 (de-militarization?). In any case, the expression "in our country" should be changed to Spain.

16. What is explained in lines 324 and following lines should be updated and quantified with the latest publication of the Spanish Agrarian Census (2020).

17. In relation to what is expressed in lines 341 and following, it would be ideal to present a map with the evolution of the main land uses.

18. The paragraph ending in line 356 needs to be supported with some quality bibliographic reference.

19. In lines 357 to 360 the expression "the cattle" is repeated. It would be advisable to lighten the wording.

20. In line 398, what does it refer to when it refers to the privatization of agricultural activity?

21. In line 409, the expression "in our country" appears again. It is more appropriate to say in Spain. A few lines below, some reference to the development of camping tourism should be included.

22. In the paragraph beginning on line 497 there is also a missing bibliographical reference.

23. Figure 9 contains expressions in Spanish that need to be translated into English.

24. What is expressed from line 538 onwards could be documented with significant photographs of the processes described.

25. In line 603 the expression "Eastern Navy" is used again; while in others it is expressed in Spanish. It is necessary to opt for the same throughout the text.

26. In the discussion and conclusions section, the validation of a hypothesis is mentioned, but this had not been expressed previously. In general, this section, rather than a summary of all the results, should present a final overall impression, point out the limitations of the work, convey the implications of the research in a broader context, demonstrate the importance of the findings and briefly reflect on new reflections and perspectives that the research could address at a later date.

Author Response

Thank you very much for your helpful comments and suggestions. I really appreciate the time and effort you have put into helping me improve the quality of the paper. I have paid close attention to these observations and have implemented your suggestions as far as possible.

It has been a great help to me, and I appreciate your effort and work in indicating point by point everything that I should improve

I have totally followed your improvement scheme, trying to make the article better.

As you indicated, I have improved the article, introducing research questions, theory and more extensive bibliography, since I have realized that it was very limited in the face of an article of this type. in addition, I have tried to support everything empirically, introducing references, hoping that the article has improved and is as good as possible.

You could find the response in the attachment.

Reviewer 4 Report

This paper introduces the urban and rural interaction with the case of Marina Oriental of Cantabria but the whole paper is heavily descriptive and most contents are rudimentary. It reads not like a journal article. 

1) There is no research question. Just describing the phenomenon with examples and simple statistic is not enough for a journal article. The dialogues or debates of your research topic should be presented in front to set up your own ACADEMIC contribution through the case. The current version is too superficial. 

2) There is no theory in this paper. The literatures cited in  this paper are limited, missing a large piece of references. Some citations need to be interpreted better, e.g., the understanding of "new urbanism" is not understood correctly and appropriately cited here. These concepts are not new either. The authors need to engage with more literatures for both theoretical and empirical ones. There are only 19 citations in total. 

3) Many paragraphs are stated without reference or empirical support. Most materials seem from authors' interpretations. The only data presented is population evolution and distribution so where those arguments on transportation accessibility, economic structure etc. are drawn from. Many numbers in the texts need to be given the sources. Now the materials is more like  a story-telling rather than making sounded arguments or insightful discussions. 

4) Th language could be modified. 

In general, my sense is that this manuscript reads like a descriptive report that lacks deep understandings of urban-rural interactions. The academic elements are missing, such as literature, theory, research questions, methodologies, and elevated discussion to state how this paper contributes to the corresponding academic community.  

Author Response

Thank you very much for your helpful comments and suggestions. I really appreciate the time and effort you have put into helping me improve the quality of the paper. I have paid close attention to these observations and have implemented your suggestions as far as possible.

It has been a great help to me and I appreciate your effort and work in indicating point by point everything that I should improve

I have totally followed your improvement scheme, trying to make the article better.

I have expanded the scale, explaining the rural-urban interaction, at the Spanish and foreign levels.

In addition, I have made changes to the structure so that its proposed results and conclusions would be clearer and separate, for the article to improve and be approved for publication.

You could find the response in the attachment.

Round 2

Reviewer 1 Report

Significant work has been carried out to inproved the original paper, following to a large extent the suggestions made by this reviewer. In partiular, the cae study has been contextualisad in the European and internatonal context, with references tu scientific texts published in high quality and hihg impact scientific journals. The structure of the article has also been improved , as well as other formaland conceptual aspects.

A typo has been detected in the bibliography. It say Pitchard, M. and should read Pitarch, M.D.

Author Response

Thank you very much again for your work and effort in correcting my article.
I am very glad that in your opinion it has improved and is more suitable.
I have made the correction in the bibliography that you indicated and I have also made small changes that other reviewers have indicated to me.
I reiterate again the thanks for your advice.

Reviewer 3 Report

The modifications made have improved the work. In my opinion it is necessary to consider the following suggestions to finalize it:

1. The summary still seems to me to be too long, it anticipates excessive information with respect to other later sections where it is more appropriate. It would be desirable to keep it to about 200 words.

2. In line 34 the word “anthropology” seems to be a bad translation of "humanización".

3. Several new paragraphs have been added to the introduction. The last two advance results from the study area. I consider that they are out of place, since the main purpose is to present the general theory based on the most updated and quality literature, in order to frame the research. Likewise, it is time to clearly state its objective. In this sense, I believe that what is stated in line 148 and following lines should be reformulated, since it is not a question of explaining the importance of this space, but of offering the keys that allow us to understand its territorial transformation. Therefore, the hypothesis would be, rather, to consider the process of expansion of the urban areas of Bilbao and Santander as the driving force behind the changes experienced in the Eastern Cantabrian Marina.

4. In the different figures that appear in the text the sources have not been included.

5. I do not see the need to include a new section called "research" (line 257). The paragraphs that compose this section seem more appropriate for the introduction.

6. The table included on page 10 seems incomplete, as it does not cover all the productive sectors nor is it broken down by municipality, a fundamental fact to explain later the internal differences of this area. I also miss references to sources, both in the table and in the text.

7. The discussion section begins by indicating that a series of conclusions will be drawn; this is not coherent. And it talks about a first hypothesis. Then, how many hypotheses does the work have? This section should be revised, because the results are not discussed with any theory, the results are simply reiterated (also in the new paragraph incorporated). In fact, no bibliographic reference appears in the whole section.

8. In the conclusions section I insist on the need to point out the limitations of the research and a possible work agenda, either to overcome these limitations or to develop new aspects not dealt with in depth (and how they could be approached), such as those pointed out in the last paragraph of this section.

9. Finally, I consider that the bibliography is still scarce; one of those that has been incorporated (number 13) contains abundant, updated and quality international references; it would be a good idea to try to incorporate some of them into the discussion section. A re-reading of this article may provide you with more ideas and appropriate frameworks to approach your research in a broader theoretical context.

Author Response

Thank you very much for your indications and for your effort and indications about my work.
I have made the corrections that you indicated to me and that I am going to list:
- First of all, I have reduced the length of the summary, since as you told me, it was too long.
- I have corrected certain words that looked like translation errors.
-I have also restructured the introduction, although I wanted to keep some details of the study area in order to understand this space before going into the details of its importance.
- I have included the sources in all the figures shown in the article.
- I have deleted the "research" section and included it in the introduction, as you indicated.
-I have restructured the table, introducing two new graphs and a new graph where all the municipalities of the study area appear (page 10).
-I have modified the discussions section, so that it does not seem that there are several hypotheses, also introducing several bibliographic references in that section.
-In the conclusions, I have pointed out the limitations of the investigation, as well as the way to solve them.
-Finally, I have expanded the bibliography with references that I have considered important and updated.

I hope that with these modifications the article will be adequate.

Again thanks for your help.